# Nutritional Status and Quality of Life: Urban–Rural Disparities and the Impact of Obesity

**DOI:** 10.3390/ijerph21111455

**Published:** 2024-10-31

**Authors:** Ane Caroline Casaes, Camilla Almeida Menezes, Ronald Alves dos Santos, Bruna Oliveira Lopes Souza, Brenda Rodrigues Brito Cunha Silva, Yuri Tabajara, Michael Nascimento Macedo, Kelvin Edson Marques de Jesus, Thainá Rodrigues de Souza Fialho, Robson da Paixão de Souza, Isadora Cristina de Siqueira, Monica Leila Portela de Santana, Ricardo Riccio Oliveira

**Affiliations:** 1Instituto Gonçalo Moniz, Fundação Oswaldo Cruz, Fiocruz Bahia, Salvador 40296-710, Brazil; ane.casaes@fiocruz.br (A.C.C.); camilla.almeida@fiocruz.br (C.A.M.); ronald.santos@fiocruz.br (R.A.d.S.); bruna.souza@fiocruz.br (B.O.L.S.); brenda.csilva@fiocruz.br (B.R.B.C.S.); yuritabajara@gmail.com (Y.T.); macedo.m@outlook.com (M.N.M.); kelvinedson2@gmail.com (K.E.M.d.J.); thaina.fialho@outlook.com (T.R.d.S.F.); robson.imuno@gmail.com (R.d.P.d.S.); isadora.siqueira@fiocruz.br (I.C.d.S.); 2Escola de Nutrição, Universidade Federal da Bahia, Salvador 40110-040, Brazil; monicalportela@gmail.com

**Keywords:** quality of life (HRQoL), nutritional status, obesity, urban–rural disparities

## Abstract

The shift in dietary habits has reshaped the population’s health profile, leading to a rise in overweight individuals and a subsequent decline in health-related quality of life (HRQoL). This study evaluated the correlations between demographic, social, and health-related factors and HRQoL in rural and urban areas of Bahia, Brazil. The cross-sectional study included 124 participants aged 18–60 who underwent interviews, anthropometric measurements, and laboratory tests. The WHOQol-BREF instrument assessed the HRQoL. The results showed that rural participants had lower educational levels, income, and access to sanitation. Despite these challenges, rural residents reported better HRQoL in psychological, social relations, and health satisfaction domains, although differences diminished after age adjustment. Urban participants, who had higher rates of obesity and related metabolic risks experienced a negative correlation between BMI and HRQoL, especially in the social relationships domain. The study highlights that environmental and social factors, such as weight-related stigma and social connections, significantly influence HRQoL in urban areas, emphasizing the need for public health interventions that address both nutritional status and urban-specific challenges.

## 1. Introduction

In recent generations, life in Brazil has changed significantly. Families have become smaller due to decreased fertility rates, infant mortality, and increased life expectancy. Brazil has also experienced rapid urbanization, leading to changes in traditional diets, known as the nutrition transition. There is a noticeable trend of replacing essential foods like beans, fruits, and vegetables with ultra-processed foods [1].

The shift in dietary habits has changed the health profile of the population, resulting in a rise in the number of overweight individuals and non-communicable diseases (NCDs). NCDs are responsible for 74% of premature deaths in Brazil. The increased prevalence of obesity is a contributing factor to the onset of NCDs and is linked to a decline in health-related quality of life (HRQoL) as well as higher healthcare costs [2].

In the 1990s, the World Health Organization (WHO) broadly defined HRQoL to include various aspects of an individual’s life, such as health. HRQoL was described as an individual’s perception of their position in life within the context of the culture and value system in which they live, as well as their goals, expectations, standards, and concerns [3,4]. Thus, HRQoL is directly related to the concept of health, defined by the WHO not only as the absence of disease but primarily as complete physical, social, and mental well-being [5].

HRQoL is a multidimensional construct influenced by various interrelated factors. Among these, demographic aspects such as age stand out since they can directly impact perception and overall well-being [6]. Social and cultural characteristics play significant roles, as cultural norms and practices can shape individual experiences and expectations of HRQoL [7]. Economic aspects, including income and access to resources, are also crucial determinants, affecting access to healthcare and the ability to meet basic needs [8]. Additionally, nutritional status and overall health are determining factors, as inadequate nutrition and pre-existing health conditions can compromise physical and psychological well-being [9,10,11].

The increasing prevalence of obesity is not only a critical public health concern but also reflects significant disparities in standards of living across various geographic and socioeconomic contexts. Recent literature highlights the complex interplay between obesity and HRQoL, particularly in populations experiencing socioeconomic disadvantages. For instance, sarcopenic obesity, a condition characterized by a detrimental fat-to-lean body mass ratio, has been linked to various chronic health conditions, including type 2 diabetes and cardiovascular diseases, adversely affecting health outcomes and HRQoL [12]. Furthermore, studies have shown that interventions aimed at promoting physical activity and nutritional education can significantly enhance health-related HRQoL in overweight and obese children, emphasizing the importance of family involvement in fostering healthier habits [13]. Additionally, systematic reviews have demonstrated that the effects of obesity phenotypes on mental health and HRQoL vary, with those exhibiting metabolic disturbances facing heightened risks [14]. These findings underscore the necessity of understanding how environmental factors and healthcare accessibility influence obesity-related disparities.

Hence, to gain a comprehensive and precise understanding of the factors impacting individual well-being, it is essential to consider these variables’ intersection in the HRQoL analysis. Consequently, this study aimed to evaluate the potential correlations between specific aspects and HRQoL among individuals living in both rural and urban areas of Bahia, Brazil.

## 2. Materials and Methods

### 2.1. Population and Data Collection

This cross-sectional study was conducted with a convenience sample of 124 participants divided into two groups: one comprising individuals residing in rural areas (n = 51) and the other comprising individuals residing in urban areas (n = 73). To ensure the statistical robustness of the study, power calculations were performed using G*Power software version 3.1.9.4, based on a medium effect size of 0.50 [15], a significance level of 0.05, and a desired power of 80% (1 − β). These calculations indicated that a sample size of 128 participants would be necessary to detect significant differences between the groups, which justifies the adequacy of the final sample of 124 participants.

Rural participants were recruited from the municipality of Conde in Bahia, 150 km from the capital, Salvador. This area is hyperendemic for schistosomiasis and has been historically monitored by this research group [16,17,18]. The individuals involved in the study were residents of the villages of Sempre Viva, Jenipapo, Camarão, and Buri, which have around 1200 inhabitants. Urban participants were recruited from two Family Health Units (FHUs) that are part of the public health system in the municipality of Salvador.

The study included individuals aged between 18 and 60 who had undergone a stool parasitological examination. Pregnant women, participants with chronic infections such as HIV and hepatitis, participants with orthopedic conditions preventing the application of the anthropometric assessment techniques outlined in the research protocol, and those unable to complete the research questionnaires were excluded from the sample.

Data collection took place in either a home setting or at the FHU through interviews and procedures conducted by trained professionals from the research team. Laboratory analyses were performed at the Laboratory of Global Health and Neglected Diseases, Gonçalo Moniz Institute, Fiocruz Bahia, and the Central Laboratory of Salvador, Bahia.

### 2.2. Demographic and Socioeconomic Evaluation

A structured questionnaire was used to gather demographic and socioeconomic information, including education level, basic sanitation, water treatment for consumption, average family income, number of people per household, and exposure to contaminated water sources.

### 2.3. Parasitological, Hematological, and Biochemical Assessment

All selected individuals underwent coproparasitological assessment. Two stool samples were collected from each participant, with the preparation and analysis of two slides from each sample using the Kato–Katz method [19,20]. Peripheral blood samples of 10 mL were collected and processed for analysis. Hematological analyses, including hemoglobin, total leukocytes, and eosinophils, were performed using the automated CELL-DYN Ruby equipment from Abbott Diagnostics^®^ (Chicago, IL, USA). To assess liver function, the markers AST and ALT were measured using the Reitman and Frankel method, and GGT was measured using the kinetic UV method.

### 2.4. Anthropometric Assessment

The participants’ anthropometric profile was assessed using the Body Mass Index (BMI), calculated using the Quetelet formula (weight in kg/height in meters^2^). Weight was measured with a digital electronic scale, brand Wiso^®^ model W920 (Nuremberg, Germany), a digital platform capable of weighing up to 200 kg with a precision of 50 g. Height was measured using a portable vertical stadiometer graduated in centimeters, with a maximum capacity of 2.10 m and a precision of 0.001 m. Both pieces of equipment were calibrated by technical assistance accredited by the National Institute of Metrology, Quality, and Technology. The BMI classification was based on the cut-off points for the adult population recommended by the WHO [21].

The nutritional assessment also considered the risk of developing metabolic complications using Waist Circumference (WC) as an indicator. WC was measured using a non-stretchable measuring tape, and the value obtained was the average of three measurements. The classification was based on the cut-off points recommended by the WHO [22].

### 2.5. Evaluation of Dietary Intake and Alcohol Consumption

The dietary profile was evaluated using a tool developed and validated by the Epidemiology Group of the Faculty of Medicine at the Federal University of Pelotas, Rio Grande do Sul. The questionnaire is based on the food groups recommended by the “10 Steps to Healthy Eating” guide from the Ministry of Health [23]. Participants were questioned about their food consumption frequency in the year leading up to the interview. They were presented with five options: less than once a week, once a week, 2 to 3 times a week, 4 to 6 times a week, and daily. The survey was conducted by a single researcher (a nutritionist) according to the guidelines outlined in the manual [23].

The investigation of alcohol intake was conducted using the instrument CAGE (Cut down drinking, Annoyed by criticism, Guilty feelings, and Eye-opener) [24,25]. To facilitate the classification of alcohol consumption levels, the study adopted the threshold values proposed. According to their methodology, consumption scores ranging from 0 to 1 are deemed to lack clinical significance, whereas scores of 2 or higher suggest the presence of alcoholism or a problematic drinking pattern.

### 2.6. Quality of Life Assessment

The study evaluated the HRQoL among the participants using the World Health Organization Quality of Life Instrument, Short Form (WHOQoL-BREF). This instrument comprises a questionnaire of 24 items systematically divided across four domains: physical, psychological, social relationships, and environment. In addition to these domain-specific items, the instrument includes two supplementary items designed to capture overarching aspects of HRQoL, specifically the individual’s Perception of Quality of Life and their Satisfaction with Health [25]. The inquiry pertains to participants’ introspective assessments concerning incidents within the fortnight leading up to the survey. Each item was evaluated through a quintet-scale metric known as the Likert scale. The outcome metrics were classified as follows: a range from 1 to 2.9 signifies a requisite enhancement in life quality; a bracket from 3.0 to 3.9 reflects a moderate life quality; scores spanning from 4.0 to 4.9 denote a superior HRQoL; and a pinnacle score of 5.0 is indicative of an exceptionally high HRQoL [26]. The inherently personal nature of the interview required that the questionnaire be administered within a confidential environment to ensure privacy.

It is important to note that the WHOQoL-BREF had to be modified using a specific equation recommended by the WHO [27]. First, for the WHOQoL-BREF analysis, questions 3, 4, and 26 were changed from negative to positive responses. Then, the average scores for each domain were calculated using an equation that required at least six questions answered for the physical domain, five for the psychological domain, two for the social domain, and six for the environmental domain. The result of this equation was multiplied by four, the number of domains in the questionnaire.

To assess HRQoL, the WHOQoL-BREF calculates a total score ranging from 0 to 100 points. A higher score indicates a better perception of HRQoL by the participant. Each of the four domains was evaluated separately ([23]). The WHOQoL-BREF was translated and validated by [28], indicating good internal consistency with a Cronbach’s alpha coefficient of 0.7, ensuring the reliability of the assessed items.

### 2.7. Statistical Analysis

Data analysis was conducted using SPSS (version 20), GraphPad Prism (version 5.0), and STATA (version 11). Descriptive statistical analysis was performed to characterize the sample and ensure the consistency of the obtained data. After confirming the normality of the numerical variables using the Kolmogorov–Smirnov and Shapiro–Wilk tests, measures of central tendency and dispersion were established. For parametric variables, means and their respective standard deviations were considered, while medians and interquartile ranges were used for non-parametric variables. Student’s *t*-test was used for parametric variables, and the Mann–Whitney test was used for non-parametric variables for comparison purposes. Categorical variables were compared using the Chi-square test and the Chi-square test for trends. Inferential statistics were performed using the Spearman correlation coefficient and linear regression model. *p* values less than 0.05 were considered significant.

## 3. Results

### 3.1. Demographic, Socioeconomic, and Clinical Assessment

Based on Table 1, the average age of participants was higher in the urban population. Most of the rural population had only completed primary education, while the majority of the urban population had completed or partially completed high school education. The entire rural population had a family income of less than one minimum wage. Regarding sanitary conditions, 33.3% of the rural population did not have access to basic sanitation, and 78.4% did not have access to treated water, in contrast to less than 6% and 11%, respectively, among urban participants. These findings confirm the socioeconomic vulnerability of the rural population compared to the urban population, which predisposes them to a higher risk of developing infectious diseases and nutritional disorders.

In this research, the parasitological evaluation found that helminth infection was more common in the rural population. All the participants from the rural areas tested positive for the parasites surveyed, with *Schistosoma mansoni* and *Trichuris trichiura* being the most widespread (66.7% and 61.9%, respectively). Coinfections were also more prevalent in the rural population, with a higher occurrence of dual infections. Interestingly, all four helminth species examined infected 7.1% of these participants. The laboratory assessment of the hepatic profile revealed that the levels of AST and GGT in the serum were significantly higher in the rural population, as well as the eosinophil count. These details can be found in Table 2.

### 3.2. Nutritional and Alcohol Consumption Evaluation

The urban population had a higher prevalence of overweight and obesity than the rural population. Among the urban participants, 66.2% were overweight, while almost half of the rural population (49%) was also overweight. As indicated by WC, it is important to highlight the high prevalence of an increased risk of metabolic complications, especially in the urban population. These details are presented in Table 3.

Regarding food consumption, most participants reported having 3 to 4 meals per day, with the urban population consuming healthier foods more often. Although no significant difference was found between the groups studied, most participants reported alcohol consumption, with a CAGE score indicating potential alcoholism or a drinking problem (see Table 3).

### 3.3. Quality of Life Assessment

The study found that, despite differences in socioeconomic status and clinical characteristics, people living in rural areas reported higher HRQoL scores in the psychological, social relations, and satisfaction with health domains. Given that the average age of the urban population was higher, an assessment was made to determine if age might be a confounding variable. After adjusting the linear regression model for age, no differences were observed in the investigated HRQoL parameters (see Table 4).

To investigate whether another factor could potentially impact the difference in quality of life (HRQoL) between the populations, we examined the relationship between HRQoL and nutritional status in each region. We found that, in the urban population, there was a negative correlation between BMI and HRQoL in the physical (r = −0.2634; *p* = 0.026) and psychological (r = −0.2456; *p* = 0.039) domains, as illustrated in Figure 1. However, this correlation was not observed in the rural population.

After this analysis, the relationship between excess weight and HRQoL became a primary focus. To ensure specificity in the findings, at this moment, the study population was limited to individuals identified with overweight and obesity. This targeted selection was consistent across both examined groups. Data analysis revealed that differences in HRQoL domains between these populations were predominantly evident in the social relation domain. Notably, participants residing in urban areas demonstrated lower scores in this category, as illustrated in Figure 2. This outcome suggests that geographical and perhaps socio-environmental factors might influence the social aspects of HRQoL in individuals with excess weight.

## 4. Discussion

The study examined the link between nutritional status and HRQoL in urban and rural areas. The key findings revealed that, in urban areas, people with excess weight had significantly lower HRQoL compared to those with normal weight. On the other hand, in rural areas, nutritional status did not considerably affect HRQoL, despite generally more adverse socioeconomic conditions. These findings suggest that the impact of nutritional status on HRQoL may be influenced by specific environmental and social factors in urban areas.

HRQoL is a topic of interest across various fields of knowledge, including philosophy, sociology, psychology, economics, and biomedicine, with a focus on individual health [24]. The increase in life expectancy and the contemporary desire to live life to the fullest have driven scientific interest in this topic [29]. The WHO developed HRQoL assessment tools to incorporate a more humanistic perspective into healthcare, beyond the traditional biomedical model focused on mortality and morbidity [30]. There was also a need to create brief yet psychometrically robust instruments to assess HRQoL comprehensively [26]. Thus, the WHO developed the WHOQoL-BREF, which assesses four domains: physical, psychological, social relations, and environment, as well as global aspects such as perception of quality of life and satisfaction with health. This was the instrument utilized in this study.

Among the urban participants assessed, the psychological and social relations domains, as well as satisfaction with health, had lower scores compared to the rural population. The psychological domain covers aspects such as enjoyment of life, finding meaning, the ability to concentrate, the perception of physical appearance, self-satisfaction, and the presence of negative feelings. The social relation domain includes personal relationships, sexual life, and support from friends. Despite the technological conveniences and networking opportunities in urban areas, these domains are not positively influenced. On the other hand, rural life may provide better indicators in these domains, even in the face of more adverse socioeconomic conditions.

One factor that could influence these results is age, as previous studies have demonstrated a negative correlation between age and HRQoL in individuals residing in rural areas [31] and between nutritional status and HRQoL in older individuals, both in terms of undernutrition and overweight [32,33,34]. In this study, the average age was higher among urban participants. This variable was identified as a confounding factor, and all subsequent analyses accounted for it statistically.

The analysis of socioeconomic and clinical characteristics, in this case defined by the presence of intestinal parasites, did not influence the HRQoL scores. Contrary to expectations, access to education and basic sanitation in rural areas is poorer, where purchasing power is lower. Where there is a higher prevalence of intestinal parasites, the psychological and social relation domains showed higher scores. A study of elderly individuals revealed that social cohesion, defined as the quality of interactions and solidarity among community members, was associated with HRQoL among those living in urban areas, but not rural areas. These results may be explained by the different social dynamics present in rural and urban lifestyles [35].

A recent study found that adults living in rural areas with higher HRQoL scores in the physical and environmental domains but lower scores in the social relation domain are more likely to develop anxiety and depressive disorders. Similarly, adults living in urban areas with lower physical scores and higher psychological scores also have an increased likelihood of experiencing these disorders [36]. In contrast, a study focusing on urban adults during the COVID-19 pandemic found that an intervention involving a rice cultivation game, designed to connect urban residents with nature, positively affected HRQoL, anxiety, depression, sustainable eating behaviors, and interpersonal relationships [37].

An additional factor that differed among the populations studied was their nutritional status. According to the Ministry of Health [2], a national survey conducted in 2019 revealed that 63% of adults in Primary Health Care were overweight (34.5%) or obese (28.5%). In the state of Bahia, these proportions were 34.3% for overweight and 24% for obesity. The participants in this study also showed a high prevalence of overweight and obesity, especially in urban areas, where there was also a higher risk of metabolic complications such as hypertension, diabetes, and cardiovascular diseases.

Regarding dietary habits of life, while no significant differences were observed between the studied groups, the majority of participants reported alcohol consumption, and their CAGE scores suggested potential alcoholism or issues related to drinking; however, it is important to note that the CAGE method has limitations, including its inability to capture the full spectrum of alcohol-related behaviors and the potential for false positives in individuals who may not meet the criteria for alcoholism.

Interestingly, in this study, both groups had access to food, with most consuming 3 to 4 meals per day. However, when comparing the intake of healthy foods, the urban population scored higher. These results suggest that access to adequate and healthy nutrition varies between populations. Despite greater access to healthy foods, other factors may be related to excess weight and HRQoL in the urban population. Consequently, an examination was conducted to determine whether nutritional status alone could impact HRQoL. This analysis showed that, apart from the psychological aspect, the physical well-being of urban individuals with overweight and obesity is also negatively related to excess weight. This correlation was not observed among rural participants.

Physical aspects include physical pain, reliance on medication or treatments, energy levels, mobility, sleep, daily activities, and work capacity. These findings suggest that, even among people with excess weight, living in rural areas positively impacts HRQoL more than in urban areas. One possible explanation for this could be the likelihood of higher levels of physical activity, which was not measured in this study. Another possible explanation for this finding may be related to the immunomodulation exerted by helminth infections such as *Schistosoma mansoni*, which was more frequent in the rural population. Studies suggest that infection with *S. mansoni* is associated with a Th2-type immune response [38,39], inhibiting the Th1-type response linked to low-grade chronic inflammation associated with obesity [40]. This scenario seems to have a protective effect against obesity [41]. Although this was not the objective of this study, further investigation is needed.

A study conducted with adults revealed positive correlations between nutritional status and all domains of HRQoL, suggesting that better nutritional status is linked to higher HRQoL [41]. Similarly, a study with overweight elderly individuals demonstrated that HRQoL was impacted, particularly in the psychological domain, with marital status being the main explanatory variable for lower scores in this domain [42].

This study showed that nutritional status significantly affected the HRQoL, especially among urban residents. This led to an examination of the subgroup of people with overweight and obesity in both rural and urban areas. The analysis confirmed that the difference in HRQoL was primarily in the domain of social relations, with lower scores among urban participants. This suggests that obesity in urban environments has a more negative impact on HRQoL. Additionally, a study of urban youth revealed that being stigmatized because of weight, rather than obesity itself, is linked to a poorer perception in all four domains of HRQoL. Individuals who experienced weight-related stigma had a more negative perception of their HRQoL compared to those who were not stigmatized [42].

The study’s results enhance our understanding of the relationship between nutritional status, HRQoL, and environmental factors. The study found that being overweight negatively impacts HRQoL in urban areas more than in rural areas. These results underscore the importance of public health policies that address not only nutritional status but also social and environmental factors unique to urban settings, such as weight-related stigma and social connections. However, the study has limitations, including the absence of the assessment of variables like physical activity and excluding qualitative aspects of individual experiences. Future research could investigate the influence of physical activity, psychological interventions, and social programs on HRQoL in different populations, especially when we consider disparities between rural and urban populations. Furthermore, exploring how implementing specific strategies to reduce weight-related stigma might enhance HRQoL in urban environments would be beneficial.

## 5. Conclusions

This study found that excess weight negatively impacts HRQoL more pronouncedly in urban areas than in rural ones, especially in the domain of social relationships among overweight urban residents. Surprisingly, nutritional status did not significantly affect HRQoL in rural areas, even though the socioeconomic conditions are more challenging. These findings indicate that environmental and social factors play important roles in mediating HRQoL.

## Figures and Tables

**Figure 1 ijerph-21-01455-f001:**
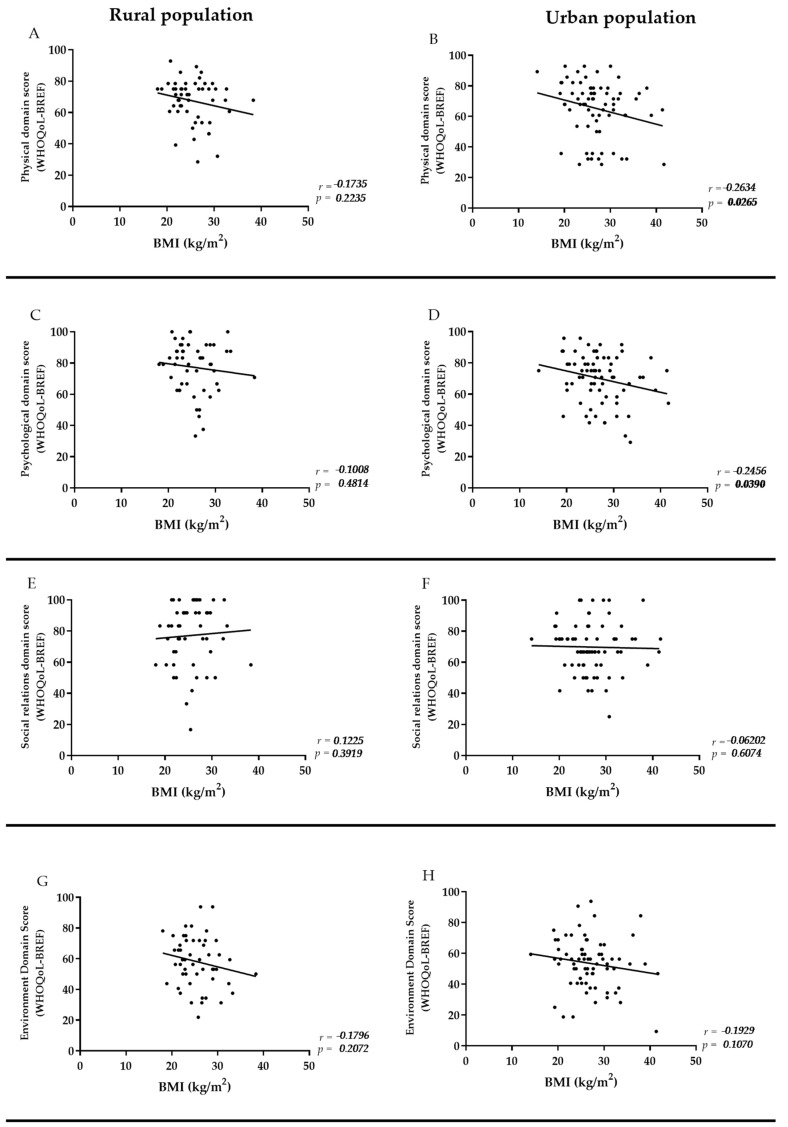
(**A**–**H**) Correlation between BMI and quality of life among rural and urban population. Statistical analysis: Spearman correlation. The *x*-axis describes the BMI values (Kg/m^2^). The *y*-axis represents the WHOQoL-BREF scores. *p* values in bold indicate statistically significant differences (<0.05).

**Figure 2 ijerph-21-01455-f002:**
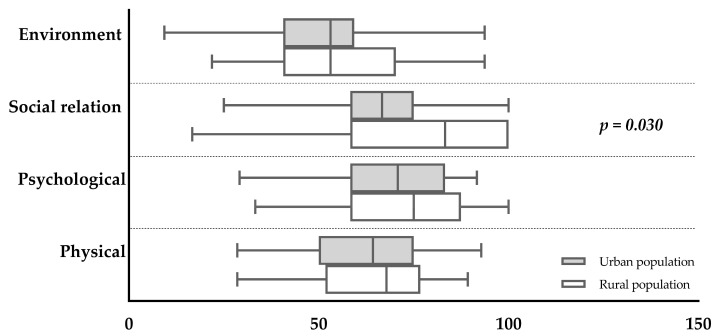
Quality of life among overweight and obese participants from rural and urban areas. The *x*-axis describes the WHOQoL-BREF scores. The *y*-axis represents the WHOQoL-BREF domains. Statistical analysis: Mann–Whitney test.

**Table 1 ijerph-21-01455-t001:** Characterization of demographic and socioeconomic factors in the study population.

Variables	Rural Population (n = 51)	Urban Population (n = 73)	*p*-Value
Age (Mean ± SD)	34.6 ± 13.0	42.6 ± 11.7	**<0.001** ^1^
Sex—Female (n [%])	29 (56.9%)	50 (68.5%)	0.185 ^2^
Level of education (n [%])			
Illiterate	5 (11.1%)	0 (0%)	**<0.001** ^2^
Elementary education	30 (66.7%)	17 (23.3%)	
Secondary education	10 (22.2%)	56 (76.7%)	
Average household income (n [%])			
Up to 1 minimum wage	49 (100%)	49 (70%)	**<0.001** ^2^
More than 1 minimum wage	0 (0%)	21 (30%)	
Sanitation at home (n [%])			
None	17 (33.3%)	4 (5.5%)	**<0.001** ^2^
Outhouse	34 (66.7%)	69 (94.5%)	
No water intake treatment (n [%])	40 (78.4%)	8 (11%)	**<0.001** ^2^

^1^ Student’s *t*-test; ^2^ Chi-square test. *p* values in bold indicate statistically significant differences (<0.05).

**Table 2 ijerph-21-01455-t002:** Comprehensive laboratory analysis of the population’s health markers.

Variables	Rural Population (n = 51)	Urban Population (n = 73)	*p*-Value
Helminth infection (n [%])			
*Schistosoma mansoni*	28 (66.7%)	0 (0%)	**<0.001** ^1^
*Ascaris lumbricoides*	18 (41.9%)	1 (6.7%)	
*Trichuris trichiura*	26 (61.9%)	0 (0%)	
Ancylostomids	9 (21.4%)	0 (0%)	
Presence of parasitic coinfections (n [%])			**<0.001** ^1^
Monoinfection	6 (14.3%)	1 (6.7%)	
Bi-infection	15 (35.7%)	0 (0%)	
Tri-infection	11 (26.2%)	0 (0%)	
Tetrainfection	3 (7.1%)	0 (0%)	
Blood markers (Mean ± SD)			
AST	24.55 ± 10.08	19.42 ± 5.88	**0.002** ^2^
ALT	20.93 ± 10.47	21.00 ± 9.34	0.975 ^2^
GGT	50.59 ± 38.00	32.51 ± 21.67	**0.004** ^2^
Hemoglobin	12.89 ± 1.30	13.33 ± 1.31	0.123 ^2^
Total leukocytes	6873 ± 1956	6601 ± 1799	0.490 ^2^
Eosinophils	571 ± 407	196 ± 213	**<0.001** ^2^

^1^ Chi-square test; ^2^ Student’s *t*-test. *p* values in bold indicate statistically significant differences (<0.05). AST: Aspartate Aminotransferase ALT: Alanine Aminotransferase; GGT: Gamma-Glutamyl Transferase.

**Table 3 ijerph-21-01455-t003:** Evaluation of dietary habits and alcohol consumption among participants.

Variables	Rural Population (n = 51)	Urban Population (n = 73)	*p*-Value
Nutritional status (n [%])			**0.028** ^2^
Underweight	1 (2.0%)	1 (1.4%)	
Normal weight	25 (49.0%)	23 (32.9%)	
Overweight	19 (37.2%)	29 (40.8%)	
Obesity	6 (11.8%)	18 (25.4%)	
Risk of metabolic complications (n [%])			**0.041** ^3^
Low to moderate	29 (56.9%)	28 (38.4%)	
High to very high	22 (43.1%)	45 (61.6%)	
Number of meals per day (n [%])			0.906 ^2^
1 or 2	2 (3.9%)	2 (3.0%)	
3 or 4	29 (56.9%)	40 (60.6%)	
5 or 6	20 (39.2%)	24 (36.4%)	
Dietary score (Mean ± SD)			
Consumption of healthy foods	2.77 ± 1.07	3.41 ± 1.31	**0.003** ^1^
Consumption of unhealthy foods	1.74 ± 0.77	1.76 ± 1.02	0.927 ^1^
Alcohol consumption (n [%])	28 (54.9%)	49 (67.1%)	0.167 ^3^
CAGE score ≥ 2	10 (43.5%)	11 (50.0%)	0.661 ^3^

^1^ Mann–Whitney U test for non-assumed equal variances; ^2^ Chi-square test for trends; ^3^ Chi-square test. *p* values in bold indicate statistically significant differences (<0.05). Nutritional status was classified according to BMI (Body Mass Index): Underweight < 18.5 kg/m^2^; Normal weight 18.5–24.9 kg/m^2^; Overweight 25.0–29.9 kg/m^2^; Obesity ≥ 30 kg/m^2^. The risk of metabolic complications was classified according to waist circumference (cm). CAGE score: Cut down drinking, Annoyed by criticism, Guilty feelings, and Eye-opener; instrument used to assess alcohol intake.

**Table 4 ijerph-21-01455-t004:** Analysis of HRQoL indicators across various domains in the population.

Variables	Rural Population (n = 51)	Urban Population (n = 73)	*p* Value ^1^	*p* Value ^2^
WHOQoL-BREF score				
Physical domain	67.44 ± 13.95	64.97 ± 18.14	0.416	0.908
Psychological domain	77.29 ± 16.25	69.81 ± 15.74	**0.** **011**	0.054
Social relations domain	77.12 ± 20.13	69.41 ± 16.32	**0.** **020**	0.078
Environment domain	58.09 ± 16.48	53.12 ± 16.40	0.101	0.146
Global score				
Quality of life perception	66.67 ± 23.80	62.39 ± 20.88	0.285	0.580
Satisfaction with health	67.65 ± 27.06	55.48 ± 24.38	**0.** **010**	0.071

^1^ Linear regression model; ^2^ Age-adjusted linear regression model. *p* values in bold indicate statistically significant differences (<0.05).

## Data Availability

The data from this study are being indexed in the Arca Dados platform and will be available for consultation upon request.

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
