# Peer review of "Nutritional Status and Quality of Life: Urban–Rural Disparities and the Impact of Obesity"

_ijerph, 2024, doi:10.3390/ijerph21111455_

Round 1

Reviewer 1 Report

Comments and Suggestions for Authors

I think that the manuscript was prepared with technical precision. It is necessary to add a precise description of the sample in the Method chapter. To accurately describe age and sex. I will recommend that it be published with minimal corrections.

Reviewer 2 Report

Comments and Suggestions for Authors

The manuscript 'Nutritional status and quality of life: urban-rural disparities and the impact of obesity' compares the significance of rural-urban disparities over changed dietary habits in Brazil. The manuscript is written clearly and precisely. Some + and - of the manuscript are:

+. QoL is a multidimensional construct, correlating each relevant dimension with QoL.

+. Part of the methodology has been explained accurately

-. Sample size determination and why the sample was collected from specific villages was not explained

-. Ordinal data was analyzed using techniques available for continuous data (Table 4)

+. Discussion was written precisely in comparing previous literature

-. Most of the results of the study are already available in previous literature

-. Recommendations are too general to the results.

The majority of the - can be improved except the second last. I recommend asking authors to improve the suggested areas before making the final decision.

Reviewer 3 Report

Comments and Suggestions for Authors

Dear Authors, 

Dear Editors,

The study offers an analysis of the relationship between nutritional status, well being, and environmental factors in urban and rural areas of Bahia, Brazil. The finding that rural participants, despite facing greater economic hardships and worse sanitation conditions, reported better quality of life scores is intriguing. This challenges the common assumption that better socioeconomic conditions automatically lead to a higher quality of life.

Areas of improvement  :

1.The sample size of 124 participants limits the generalizability of the findings. Expanding the sample size and including more regions could strengthen the conclusions.

2.It would be beneficial to further discuss existing literature on standards of living disparities related to obesity in other geographic or socioeconomic contexts. This would help to justify the research focus more explicitly.

3.Thestudy uses BMI and waist circumference as indicators of nutritional status. Given that BMI has limitations in distinguishing between fat and muscle mass, including additional body composition measures (like body fat percentage) would enhance the validity of the findings.

Discussing the potential limitations of CAGE score or considering a more specific measure for assessing alcohol-related behaviors could be beneficial

4.While the authors adjust for age, further explanation of how confounding variables like socioeconomic status, education, and physical activity (which is mentioned but not measured) were controlled for would strengthen the analysis.

5.The observation that urban participants with excess weight report lower quality of life in psychological and social domains is compelling. However, further explanation is needed on the mechanisms behind these findings, possibly incorporating more detail on weight stigma or societal pressures in urban environments.

6.Given its known impact on life quality and obesity, the omission of physical activit data is a significant limitation. A stronger rationale for why this was not included should be provided, and a recommendation for future research in this direction could be done. Given that physical activity levels are likely higher in rural areas, this might explain some of the differences observed between the populations.

7.The potential immunomodulatory role of helminth infections is interesting but underdeveloped. More discussion on the evidence supporting this hypothesis, along with any direct study findings is needed.

In conclusion, while it has some limitations, I think that the findings of  the study are valuable for informing public health policies in both urban and rural contexts.
